# A Novel Lateral Flow Immunochromatographic Assay for Rapid and Simultaneous Detection of Aflatoxin B1 and Zearalenone in Food and Feed Samples Based on Highly Sensitive and Specific Monoclonal Antibodies

**DOI:** 10.3390/toxins14090615

**Published:** 2022-09-02

**Authors:** Yanan Wang, Xiaofei Wang, Shuyun Wang, Hanna Fotina, Ziliang Wang

**Affiliations:** 1Henan Institute of Science and Technology, College of Animal Science and Veterinary Medicine, Xinxiang 453003, China; 2Faculty of Veterinary Medicine, Sumy National Agrarian University, 40021 Sumy, Ukraine; 3Xinxiang Institute of Engineering, College of Bioengineering Henan, Xinxiang 453003, China

**Keywords:** mycotoxins, highly sensitive and specific monoclonal antibodies, dual lateral flow immunochromatographic assay, immunoassay, agro-products

## Abstract

Simultaneous aflatoxin (AFB1) and zearalenone (ZEN) contamination in agro-products have become widespread globally and have a toxic superposition effect. In the present study, we describe a highly sensitive and specific dual lateral flow immunochromatographic assay (dual test strip) for rapid and simultaneous detection of AFB1 and ZEN in food and feed samples based on respective monoclonal antibodies (mAbs). Two immunogens AFB1-BSA (an AFB1 and bovine serum albumin (BSA) conjugate) and ZEN-BSA (a ZEN and BSA conjugate) were synthesized in oximation active ester (OAE) and amino glutaraldehyde (AGA). The molecular binding ratio of AFB1:BSA was 8.64:1, and that of ZEN:BSA was 17.2:1, identified by high-resolution mass spectrometry (HRMS) and an ultraviolet spectrometer (UV). The hybridoma cell lines 2A11, 2F6, and 3G2 for AFB1 and 2B6, 4D9 for ZEN were filtered by an indirect non-competitive enzyme-linked immunosorbent assay (inELISA) and an indirect competitive enzyme-linked immunosorbent assay (icELISA), respectively. As AFB1 mAb 2A11 and ZEN mAb 2B6 had the lowest 50% inhibitive concentration (IC50) and cross-reactivity (CR), they were selected for subsequent experiments. By systematically optimizing the preparation condition of gold nanoparticles (AuNPs), AuNPs-labeled mAbs, and detection condition, the visual limit of detection (LOD) of the dual test strip was 1.0 μg/L for AFB1 and 5.0 μg/L for ZEN, whereas that of the test strip reader was 0.23 μg/L for AFB1 and 1.53 μg/L for ZEN. The high reproducibility and stability of the dual test were verified using mycotoxin-spiked samples. The dual test strips were highly specific and sensitive for AFB1 and ZEN, which were validated using liquid chromatography-tandem mass spectrometry (LC-MS/MS). Thus, the proposed AFB1 and ZEN dual test strip is suitable for rapid and simultaneous detection of AFB1 and ZEN contamination in food and feed samples.

## 1. Introduction

Mycotoxins are low-molecular-weight, toxic secondary metabolites produced under particular environmental conditions by various fungi such as *Fusarium*, *Aspergillus*, and *Penicillium*, and contaminate a large variety of agro-products and feed worldwide, which can cause the reduction in agriculture production and livestock production, pose a great threat to human and animal health through the food and feed, and result in the tremendous economic losses [1,2]. Particularly in recent years, the co-occurrence of multiple mycotoxins in agricultural products has been increasingly frequent, and their co-existence displays synergistic and additive toxicological effects in humans or animals [3,4]. Thus, a single-target mycotoxin detection method cannot meet the actual needs of the food industry and feed industry, and a method to determine multiple mycotoxins needs to be established urgently to monitor their co-contamination.

So far, about 400 mycotoxins have been reported, which aflatoxin B1 (AFB1) and zearalenone (ZEN) are frequently and widely present in agricultural products, and are also the targets to monitor in this study. Aflatoxins (AFs) produced under natural conditions mainly include AFB1, AFB2, AFG1, and AFG2, and possess carcinogenicity, mutagenicity, malformation, neurotoxicity, and immunotoxicity properties in humans and animals [5], among which AFB1 is the most common and toxic aflatoxin, grouped as a class I carcinogen by the International Organization for Research on Cancer (IARC) [6]. Many countries have proposed specific maximum permitted levels (MPLs) for AFB1 in food [7]. For instance, the MPLs of AFB1 in cereals are capped at 20 μg/kg in China and America and 2 μg/kg in the European Union (EU) [8]. Zearalenone (ZEN) and its metabolites (ZENs), including α-zearalenol (α-ZOL), β-zearalenol (β-ZOL), α-zearalanol (α-ZAL), β-zearalanol (β-ZAL) and zearalanone (ZON)), are non-steroidal estrogenic compounds that mainly disrupt estrogen secretion, disturb the reproductive system, and display estrogenic effects in humans and animals. Furthermore, ZENs are genotoxic, immunotoxic, and carcinogenic compounds with endocrine toxicity properties [9]. As ZEN is the most common, abundant, and toxic mycotoxin, it has become the main target for food quality and safety monitoring globally. The MPLs of ZEN is set 75 μg/kg in cereals and 100 μg/kg in feeds in the UK [10], 100 μg/kg in cereals in Italy [11], and 50 μg/kg in cereals in Australia [12]. Owing to their extreme toxicity and universal co-existence, the establishment of rapid and simultaneous detection methods for AFB1 and ZEN is vital in agro-food and feed safety control.

Currently, various methods for multiple mycotoxins determination have been well-developed, including instrumental analysis and immunoassays. Instrumental multi-analysis methods mainly comprise high-performance liquid chromatography (HPLC) [13] and liquid chromatography-tandem mass spectrometry (LC-MS/MS) [14]. Despite their excellent sensitivity and accuracy, these methods are complex, require sophisticated equipment and a highly trained workforce, and are expensive. Thus, these methods are only suitable for testing a few samples and not appropriate for the on-site monitoring of many samples [15]. As the food industry and feed industry require more rapid and cost-effective methods for multiple mycotoxins detection, several immunoassays such as enzyme linked immunosorbent assay (ELISA) [16] and immunofluorescence assay (IFA) [17] have been developed for commercial application in recent years, but these methods also require some complicated operations and need to be in the detection process for a long time. Fortunately, the lateral flow immunoassay (LFIA, also called immunochromatographic assay (ICA)), and especially the gold immunochromatography assay (GICA, also known as test strip), are examples of the most widely used rapid detection technology globally [18]. With the advantages of being rapid, simple, portable, inexpensive, on-site testing, and high throughput property, GICA is the most widely used technique for field food safety analysis [19]. Shim et al. [20] established a dual test strip based on the self-made ochratoxin A (OTA) monoclonal antibody (mAb) and ZEN mAb. The visual limit of detection (LOD) of this technique for OTA and ZEN were 2.5 μg/kg and 5.0 μg/kg, respectively. However, the specificity of the method for both OTA and ZEN was low, showing cross-reactivity (CR) with their respective homologues. Chen et al. [21] developed a triple test strip for simultaneous detection of AFB1, ZEN, and OTA using self-made AFB1 mAb, ZEN mAb, and OTA mb, respectively. The visual LOD of the technique for AFB1, ZEN, and OTA were 10 μg/kg, 50 μg/kg, and 15 μg/kg, respectively, which is generally low for their respective sensitivity.

Notably, high-quality monoclonal antibodies (mAbs) are essential to immunoassays, since their efficacy depends on the specificity and sensitivity of the mAbs. In recent years, some scholars have achieved good progress in developing AFB1 mAb and ZEN mAb with high specificity and high sensitivity. Our research on the synthesis of AFB1 immunogens and the characteristics of corresponding antibodies showed that among the six immunogen synthesis methods, including oximation active ester (OAE), methylation of ammonia (MOA), mixed anhydride (MA), semi acetal (SA), epoxide (EP), and enol ether derivative (EED), OAE was the best method for preparing highly specific AFB1 antibodies [22], consistent with Li et al. findings [23]. We conducted similar research on ZEN and found that amino glutaraldehyde (AGA) was better than oxime active ester (OAE), formaldehyde (FA), 1,4-butanediol diglycidyl ether (BDE) in preparing highly specific ZEN antibodies [24], consistent with Teshima et al. findings [25]. To the best of our knowledge, there is no report on preparing highly specific AFB1 and ZEN mAbs and a corresponding dual test strip.

Herein, we developed a rapid and sensitive dual test strip method for the simultaneous detection of AFB1 and ZEN based on their specific and sensitive mAbs and gold nanoparticles (AuNPs). By systematically optimizing, the method for simultaneous detection of AFB1 and ZEN was rapid, simple, portable, and cheap, and can be used for AFB1 and ZEN determination in real agro-foods, feed samples. Its feasibility was validated using LC-MS/MS.

## 2. Results and Discussion

### 2.1. Characterization of Mycotoxins and Carrier-Protein Conjugates

AFB1 and ZEN are small molecules that have only reactivity without immunogenicity. Haptens can only combine with macromolecular protein to form an immunogenic molecule that induces the production of specific antibodies. The specificity of a hapten antibody mainly depends on the recognition of the characteristic structure of the hapten on the immunogen by B cells. Owing to the selection of different active sites and the introduction of different spacer arms in the synthesis of immunogens, the spatial conformation of the immunogen molecule, the exposure degree, and the degree of recognition and memory by B cells differ. Thus, the specificity of the antibodies secreted by these memory B cells after maturation into plasma cells is also different [26]. Therefore, the method of immunogen synthesis is crucial for the preparation of highly specific antibodies. Our previous study analyzing six types of AFB1 and five types of ZEN immunogen synthesis methods revealed that OAE and AGA were the best methods for preparing AFB1 and ZEN-specific antibodies, respectively [22,24]. In the current study, AFB1 was haptenized by introducing an active carboxyl group to the C1 position carbonyl group on the AFB1 molecule through the oximation reaction. AFB1-BSA immunogen (AFB1 and bovine serum albumin (BSA)) conjugate were synthesized using the OAE method. Similarly, a reactive amino group was conjugated onto benzene ring-containing C5 position of ZEN through nitration and reduction in the ZEN molecule. The ZEN-BSA immunogen was synthesized using the AGA method.

High-resolution mass spectrometer (HRMS) data demonstrated that AFB1 and ZEN were successfully haptenized. Quantitative analysis of AFB1-BSA and ZEN-BSA conjugates was performed using an ultraviolet spectrometer (UV) (Figure 1a) and ZEN-BSA (Figure 1b). For AFB1-BSA, in the range of 220 to 440 nm, the characteristic absorption peak of BSA is 278 nm, and the characteristic peak of AFB1 is 363 nm, while AFB1-BSA synthesized by the OAE method contains the characteristic peaks of BSA and AFB1, indicating successful AFB1-BSA synthesis. For ZEN-BSA, in the range of 220 to 440 nm, BSA has a characteristic peak at 278 nm, whereas ZEN has characteristic peaks at 236, 274, and 316 nm. ZEN-BSA prepared using the AGA method displayed the characteristic absorption peaks of both BSA and ZEN, indicating successful ZEN-BSA. The calculated molar ratio of AFB1 to BSA and ZEN to BSA based on the Lambert–Beer law were 8.64:1 and 17.2:1, respectively.

### 2.2. Preparation and Assessment of AFB1 mAbs and ZEN mAbs

The specificity of antibodies depends on not only the physicochemical properties of the immunogen but also the immunization method, route of administration, dose, challenge interval, presence or absence of adjuvant, immunization times, and individual differences with the immunization dose and time interval being the most critical. Li et al. [23] hypothesized that small doses of immunogen could induce the generation of antibodies with a narrow recognition spectrum, suitable for preparing highly specific antibodies. In contrast, large immunogen doses could induce the generation of antibodies with a broad recognition spectrum, suitable for preparing broad-spectrum class-specific antibodies. Neuberger et al. [27] and Gilfillan et al. [28] reported that a specific immunization interval could improve the affinity and specificity of antibodies, and the immunization interval should generally not be less than two weeks. Therefore, we adopted a small dose (30 μg/mL), a longer immunization interval (four weeks), multiple immunization sites (four to six) on the back, and multiple times (five times) immunization methods.

The polyclonal antibodies (pAbs) titers against AFB1 and ZEN in mice on day 25 after vaccination were detected using a homologous indirect non-competitive enzyme-linked immunosorbent assay (inELISA). The 50% inhibitive concentration (IC50) and the CR% levels of the pAbs were detected using a homologous indirect competitive enzyme-linked immunosorbent assay (icELISA). For the AFB1-BSA (OAE) group, the highest titer of 1:(3.2 × 10^3^) (Figure 2a), the lowest IC50 value of 21.12 μg/L (Figure 2b), and the smallest CR% with the AFB1 analogs (Figure 2c) were observed in mouse 2. For the ZEN-BSA (AGA) group, the highest titer of 1:(6.4 × 10^3^) (Figure 3a), the IC50 of 18.77 μg/L (Figure 3b), and the smallest CR% with the ZEN analogs (Figure 3c) were observed in mouse 5. Thus, mouse 2 in the AFB1-BSA (OAE) group and mouse 5 in the ZEN-BSA (AGA) group were used in subsequent cell fusion experiments.

B lymphocytes recognize, remember, and produce antibodies against a protein, spacer arm, and hapten for hapten-protein conjugates. Although a single B lymphocyte can only produce one specific type of antibody, different types and subclasses of antibodies show varied sensitivities and specificities to the same determinants [29]. Therefore, screening for positive hybridoma cell lines is also a key step that determines the sensitivity and specificity of mAbs. In this study, a gradient two-step screening method was adopted. In the initial screening of hybridoma cells twelve to fourteen days after cell fusion, high concentrations of coating antigens (5 μg/mL AFB1-OVA or ZEN-OVA) and inhibitors (1000 μg/L AFB1 or ZEN) were used in inELISA and icELISA to avoid missing cells that secrete high-affinity antibodies but are not dominant in the culture plate. In the second screening of single hybridoma cells after thrice subclone, low concentrations of coated antigens (0.5 μg/mL AFB1-OVA or ZEN-OVA) and low concentrations of inhibitors (10 μg/L AFB1 or ZEN) were used in the inELISA and icELISA assays to generate the desired high-affinity mAbs.

After cell fusion culture and filtration, the positive hybridoma cell lines 2A11, 2F6, and 3G2 derived from AFB1-BSA (OAE) and 2B6 and 4D9 derived from ZEN-BSA (AGA) were screened out. Under the optimal icELISA conditions (type of coating antigen, optimal antigen, antibody, GaMIgG-HRP, and organic solvent concentrations, the ionic strength, and the pH value), the sensitivities (IC50) and CRs of AFB1 mAbs (Table 1) and ZEN mAbs (Table 2) were determined using icELISA. As shown in Table 1, the IC50 of AFB1 mAb 2A11 (6.28 μg/L) was lower than that of 2F6 (7.85 μg/L) and 3G2 (14.36 μg/L) by 25% (*p* < 0.05) and 128.66% (*p* < 0.05), respectively, and the CRs against AFB1 analogs of 2A11 (0.91–4.35%, mean 2.52%) was lower than that of 2F6 (0.98–4.65%, mean 2.72%) and 3G2 (0.96–14.36%, mean 3.15%) by 7.94% (*p* < 0.05) and 25% (*p* < 0.05), respectively. The lowest IC50 and CR were observed for mAb 2A11. Accordingly, mAb 2A11 was selected for the subsequent studies analyses. Likewise, the IC50 of ZEN mAb 2B6 (10.38 μg/L) was lower than that of 4D9 (17.23 μg/L) by 65.99% (*p* < 0.05), and the CRs against ZEN analogs of 2B6 (1.28–4.27%, mean 2.308%) was lower than that of 4D9 (1.35–4.88%, mean 2.546%) by 10.31% (*p* < 0.05). Therefore, mAb 2B6 was selected for further experiments (Table 2).

Additionally, the AFB1 mAb 2A11 obtained in this study was compared with previously reported AFB1-specific mAbs (Table 3). The mAb 2A11 was highly sensitive, with an IC50 (AFB1) of 6.28 μg/L. Its specificity was very high, with the CR reaching (AFB2) of 4.35%, in line with the expected research goal. Regarding specificity, the highest CR of mAb 2A11 was similar to those of mAb 1B5 and mAb 2F12 reported by Li et al. [23] but superior to other AFB1 mAbs. For sensitivity, the IC50 of mAb 2A11 was higher than those of mAb 1B5 and mAb 2F12 reported by Li et al. [23], mAb 1F7 reported by Jiang et al. [30], mAb 34 reported by Kolosova et al. [31], and mAb 3A12 reported by Zhang et al. [32]. Therefore, the sensitivity of mAb 2A11 needs further improvement.

Similarly, the sensitivity of ZEN mAb 2B6 was compared with other previously reported ZEN-specific mAbs (Table 4). The IC50 of mAb 2B6 was 10.38 μg/L, whereas its CR was 1.35–4.88 μg/L (ZEN analogs). Therefore, mAb 2B6 is highly sensitive and specific for ZEN, in line with the study aim. Regarding sensitivity, the IC50 of mAb 2B6 was lower than that of mAb 7-1-14 secreted using the same AGA method as reported by Teshima et al. [25] and Gao et al. [33] but was higher than that of mAb 4A3 [34], and mAb 2D8 [35]. For its specificity, the CR value of ZEN mAb 2B6 was higher than that of mAb # and mAb 7-1-144. Therefore, the specificity of mAb 2B6 needed to be further improved.

### 2.3. Identification of AuNPs

The UV and transmission electron microscope (TEM) results for AuNPs are shown in Figure 4. The maximum absorption peak (λmax) of AuNPs was at 523.8 nm, and the maximum absorption value (Amax) was 0.941. The absorption peak of AuNPs was only one and small. According to Cvak et al. [36] findings, AuNPs have uniform size and distribution. The particles are 25 nm wide (Figure 4a). TEM revealed that the AuNPs were uniformly distributed and with a regular shape. The average size of 100 randomly selected sample particles was 25 ± 1.0 nm (Figure 4b), consistent with the UV scanning results.

### 2.4. The Optimal pH and the of AuNPs for Labeling mAb

The pH directly affected the binding of AuNPs-labeled mAb. Under optimal pH, AuNPs adsorbs onto mAb to saturation, and the absorbance value of AuNPs-labeled mAb was the highest. At low pH, AuNPs precipitated due to self-aggregation. At high pH, only a small amount of mAbs was adsorbed by AuNPs, the test strip color was faint, and the sensitivity of the test strip was reduced. UV analysis for the optimal pH level for the absorption of AuNPs on mAb is shown in Figure 5. The adsorption was highest at pH 6.5, almost reaching the saturation point. Therefore, the optimal binding pH between AuNPs and AFB1 mAbs was 6.5 (Figure 5a). Similarly, the optimal pH for binding between AuNPs and ZEN mAb was 7.0 (Figure 5b). UV analysis for the optimal amount of AuNPs and mAb is shown in Figure 6. The results showed that the absorbance was highest when the concentration of AFB1 mAb was 6.25 μg/mL. The optimal concentration of AFB1 mAb was 7.0 μg/mL (6.25 × 110% ≈ 7.0) (Figure 6a). The sensitivity of AFB1 mAbs could reduce at very high concentrations. Figure 6b shows the optimal canceration of ZEN mAbs (3.5 μg/mL).

### 2.5. The Optimal Technical Parameters of the Dual Test Strip

The optimal working combination concentration between AuNPs-labeled AFB1/ZEN mAb and coating antigen AFB1-BSA/ZEN-BSA determined using the chessboard method is shown in Table 5. In particular, the optimal working concentrations for AuNPs-labeled AFB1 mAb and AFB1-BSA were 1:4 and 1.0 mg/mL, respectively, and those of AuNPs-labeled ZEN mAb and ZEN-BSA were 1:4 and 2.0 mg/mL, respectively. The optimal combination for AuNPs-labeled AFB1/ZEN mAb and the coating antigen AFB1-BSA/ZEN-BSA are shown in Table 6. The optimal combination type for AuNPs-labeled AFB1 mAb and the coating antigen was AFB1-BSA (OAE) and ZEN-BSA (AGA) for AuNPs-labeled ZEN mAbs.

The test results of six nitrocellulose (NC) membranes and four conjugate pads are shown in Figure 7. As shown in Figure 7a, red bands were observed for all the NC membranes. In contrast, the color of Millipore 135 membrane was clearer at 10 min. Therefore, Millipore 135 membrane was selected as NC. Meanwhile, as shown in Figure 7b, the conjugate pads can completely release the AuNPs-labeled mAb within 10 min without background color, but in contrast, 8964 is much more obvious and clear. Thus, 8964 was selected as the conjugate and sample pad in this study. The following buffers were used in our experiment: PBS (0.01 M, pH7.4, comprising NaCl 137 mM, Na_2_HPO_4_▪12H_2_O 10 mM, KCl 2.68 mM, and KH_2_PO_4_ 1.47 mM) was the coating antigen dilution buffer; BBS (0.05 M pH 7.4) supplemented with 2.0% of BSA, 4.0% of sucrose, and 0.05% of NaN_3_ was the dilution buffer for AuNPs-labeled mAbs; PBS (0.01 M, pH7.4) supplemented with 2.0% of BSA, 0.5% of Tween-20, and 0.05% of NaN_3_ was the NC membrane blocking buffer. BBS (0.05 M pH 7.4) supplemented with 2.0% of BSA, 4.0% of sucrose, 0.5% Tween-20, and 0.05% NaN_3_ was the conjugate and sample pad processing buffer.

### 2.6. Validity of the Dual Test Strip Test

Sensitivity is one of the key indicators for an accurate immunoassay, represented by LOD. The LOD of the dual test strip method is shown in Figure 8. The T1 line represents AFB1 ≥ 1.0 μg/L, whereas the T2 line represents ZEN ≥ 5.0 μg/L. T1 and T2 lines disappeared completely, and the sample was positive. Therefore, the visual LOD of AFB1 and ZEN of the dual test strip test was 1.0 μg/L and 5.0 μg/L, respectively. The test results using a test strip reader are shown in Table 7. The respective standards are shown in Figure 9. The linear regression equation of AFB1 was y = −52.526x + 46.762, the IC50 was 1.15 μg/L, and the LOD was 0.23 μg/L. Meanwhile, the linear regression equation of ZEN was y = −34.215x + 73.644, the IC50 was 4.91 μg/L, and the LOD was 1.53 μg/L.

The repeatability analysis using AFB1/ZEN spiked samples revealed that the test was reproducible (Table 8).

The validity period of the results is shown in Table 9. The appearance and sensitivity of the dual test strip had not changed in six months at 4 °C and 25 °C. The T1 and T2 lines are clearly visible. No false positives and false negatives were detected. The results remained unchanged when the strip was stored at 37 °C for four months, 4 °C, and 25 °C for six months. However, the color of the T1 line T2 lines faded when the strip was stored for over four months (150 days), giving false-negative results. On day 180, the color of lines T1 and T2 faded, giving false negatives. Therefore, the dual test strip can be stored in the refrigerator (4 °C), and room temperature (25 °C) for six months, and at high temperature (37 °C) for four months.

The test results for positive corn samples using the dual test strip are shown in Table 10. AFB1 was detected in 12 of the 20 positive samples by dual test strip; the positive value was ≥1.0 μg/L, the positive value detected by LC-MS/MS was 1.0–11.9 μg/L, and their positive coincidence rate was 100%. Similarly, ZEN was detected in eight of the 20 positive samples; the positive value was ≥5.0 μg/L, the positive value detected by LC-MS/MS was 5.0–367.6 μg/L, and their positive coincidence rate was 100%. The test results of natural samples are shown in Table 11. A total of 60 samples were analyzed. Of these, 39 positive samples were correctly detected, and the positive rate was 65%. All the positive samples were correctly detected, and the detection rate was comparable to that of LC-MS/MS. Of the 39 positive samples, 22 samples were positive for AFB1. The positive value of the dual test strip was ≥1.0 μg/L, whereas that of LC-MS/MS was 1.0–14.3 μg/L. Furthermore, 17 samples were positive for ZEN, the positive value of the dual test strip was ≥5.0 μg/L, whereas that of LC-MS/MS was 5.0–510.6 μg/L, and the positive agreement rate of the two tests was 100%.

Additionally, although the developed dual test strip method has obvious advantages, it still has the following three limitations. First, its accurate quantification cannot be achieved. Since its detection results are qualitatively and semi-quantitatively evaluated by observing the color changes produced by the accumulation of AuNPs using naked eyes, its results cannot be accurately quantified [37]. Second, its sensitivity needs to be improved. The amplification effect of GICA on the antigen-antibody reaction signal is not as good as that of ELISA and IFA, and there is a certain missed detection rate for weakly positive samples in the naked eye observation test results [38]. Third, it is difficult to choose a sample processing method. Different targets have different solubility, different sample processing methods are required, and the obtained sample processing solutions have different compatibility with the dual test strip buffer system, which may easily lead to poor stability of the detection results [39]. AFB1 is the most toxic toxin in AFs, and its median lethal dose (LD50) is 6.5–16.5 mg/kg [40]. The toxicity of ZEN is relatively low, with an LD50 of 2000–20,000 mg/kg in mice [41]. The most stringent MPLs for AFB1 and ZEN in food and feed are 2 μg/kg [8] and 50 μg/kg [12], respectively, while the visual LODs of AFB1 and ZEN by the proposed method are 1.0 μg/L and 5.0 μg/L, respectively. Therefore, the proposed method can meet the actual detection needs. Moreover, there have been previous reports on the use of GICA for the detection of AFB1 and ZEN. Chen et al. [21] developed a triple test strip for the detection of AFB1, ZEN and ochratoxin A (OTA); their LODs were 10 μg/L, 50 μg/L and 15 μg/L, respectively, due to the large CR of the mAbs used, and the method was mainly used for the total detection of the target and its homologues. Song et al. [42] established the same method for detection of AFB1, ZEN and deoxynivalenol (DON), but also for total detection of the target and its homologues. In view of this, drawing on the methods of Li et al. [23] and Teshima et al. [25]., this study prepared highly specific and sensitive mAbs for AFB1 and ZEN, established a double test strip method, and achieved simultaneous, specific, sensitive and rapid determination of AFB1 and ZEN.

## 3. Materials and Methods

### 3.1. Chemicals and Materials

The mycotoxins, including AFB1, AFB2, AFG1, AFG2, ZEN, α-ZOL, β-ZOL, α-ZAL, β-ZAL, ZON, deoxynivalenol (DON), fumonisin B1 (FB1), ochratoxin A (OTA), and chemical reagents, including carboxymethoxylamine hemihydrochloride (CMO), N-hydroxysuccinimide (NHS), N-(3-dimethylaminopropyl)-N’-ethyl-carbodiimide (EDC), and Tween-20 were purchased from Sigma Chemical Co. (St. Louis, MO, USA). BSA, ovalbumin (OVA), Freund’s complete adjuvant (FCA), Freund’s incomplete adjuvant (FIA), RPMI-1640 with L-glutamine, horseradish peroxidase (GaMIgG-HRP)-conjugated goat anti-mouse IgG, polyethylene glycol 1500 (PEG 1500, 50%), hypoxanthine aminopterin thymidine (HAT), and hypoxanthine thymidine (HT) were purchased from Pierce Biotechnology, Inc. (Rockford, IL, USA). Formaldehyde (FA), glutaraldehyde (GA), ethylenediamine (EDA), and dimethyl sulphoxide (DMSO) were purchased from J&K Chemicals Ltd. (Shanghai, China). Chloroauric acid, trisodium citrate, sodium azide, and boric acid were purchased from China National Pharmaceutical Group Corporation (Beijing, China). The chemicals and organic solvents used in the experiments were of analytical grade and, therefore, did not require further purification. The NC membrane, sample and conjugate pads (glass fiber), absorbent pad, and adhesive backing card were purchased from Millipore Corporation (Millipore, Bedford, MA, USA). Seven-week-old female Balb/c mice (license number of SCXK (YU) 2015-0004) with an average weight of 18.2 ± 0.5 g were purchased from the Laboratory Animal Center of Zhengzhou University (Zhengzhou, China) and were reared under controlled conditions in our laboratory. Murine myeloma NS0 cells were purchased from the Key Laboratory of Animal Immunology of the Ministry of Agriculture (Zhengzhou, China) and were cultured in RPMI-1640 medium supplemented with 20% (*v*/*v*) fetal bovine serum in our laboratory.

### 3.2. Instrumentations

The microplates were analyzed using a Multiskan MK3 microplate spectrophotometric reader (Thermo, Waltham, MA, USA). UV spectra of the mycotoxins were acquired using a DU-800 UV-visible spectrophotometer (Beckman Coulter Inc., Fullerton, CA, USA). Haptenization of AFB1 and ZEN was verified using a hybrid quadrupole-time of flight mass spectrometer (Q/TOF, HRMS; SYNAPT HDMS, Waters, UK). An XYZ-3210 3D spray point platform (BioDot Inc., Irvine, CA, USA) and a CM 4000 guillotine-cutting module (Kinbio Tech Co., Ltd., Shanghai, China) were used to analyze the test strip assembly. The size, distribution, and morphology of the AuNPs were analyzed using an H-7650 transmission electron microscope (Hitachi Limited, Tokyo, Japan). The coloration signals were detected using a BioDot TSR 3000 test strip reader (Gene Company Limited, Shanghai Branch, Shanghai, China), whereas an LC-MS/MS-8030 reader equipped with an electrospray ionization interface (Shimadzu, Kyoto, Japan) was used to validate the test results.

### 3.3. Synthesis and Verification of Mycotoxin-Protein Conjugates

AFB1-BSA synthesis was performed using the OAE method as previously described by Kolosova et al. [31], but with slight modifications. Briefly, AFB1 was haptenized. Then, 5 mg (0.02 mM) of AFB1 and 12 mg (0.06 mM) of CMO were dissolved in 5 mL of pyridine and incubated at 37 °C in the dark for 24 h with continuous string. CMO (12 mg, 0.06 mM) was added to the mixture before further incubation for 24 h under the described conditions. The mixture was freeze-dried for 24 h to obtain a white powdery compound, the target product of AFB1 oxime (AFB1O), which was verified using HRMS. HRMS (ESI) m/z calculated for C_19_H_15_NO_8_ [AFB1O + H^+^] 385.3243, found 385.3237. To synthesize AFB1-BSA, 5.8 mg (0.015 mM) of AFB1O was dissolved in 1.0 mL of dimethylformamide (DMF) before adding 3.1 mg (0.016 mM) of EDC. The mixture was incubated at room temperature in the dark for 2 h with contiguous stirring to obtain the hapten activation solution. Thereafter, 20 mg (0.0003 mM) of BSA was dissolved in 1.0 mL PBS (0.01 M, pH 7.4). The hapten activation solution was then added dropwise to the BSA solution with continued stirring for 2 h at room temperature. The reaction mixture was dialyzed with PBS for three days at 4 °C. The resulting AFB1O-BSA conjugate was stored at 4 °C until further use. The schematic flow for AFB1-BSA (OAE) synthesis is shown in Figure 10.

ZEN-BSA was synthesized using the AGA method as previously described by Teshima et al. [25], but with minor modifications. Briefly, ZEN was first haptenized. 31.84 mg (0.1 mM) of ZEN, 23.33 mg (0.1 mM) of ZrO (NO_3_)_2_, and 5 mL of acetonitrile were added to a reaction tube under a nitrogen atmosphere. The mixture was stirred under a refluxed condition for 16 h. The mixture was then filtrated and distilled under low pressure. The crude product was purified using column chromatography. A yellow solid, an intermediate product of 5-NO_2_-ZEN, was obtained. Iron powder was added to 109 mg (0.3 mM) of 5-NO_2_-ZEN dissolved in 5 mL of hydrochloric acid. The experiment was performed at room temperature for 2 h. The solution was filtered, concentrated in a vacuum, and dissolved in 5 mL of double-distilled water. The pH was adjusted to 7.0 using K_2_CO_3_. The above solution was extracted with dichloromethane (DCM, 3 × 10 mL). The organic layer was washed off using salt water, dried with anhydrous Na_2_SO_4_, and then concentrated in a vacuum to obtain the light yellow solid (5-NH_2_-ZEN), verified using HRMS. HRMS (ESI) m/z calculated for C18H24NO5 [ZEN-NH_2_ + H^+^] 334.1649, found 334.1643. Subsequently, 5 mg (0.015 mM) of 5-NH_2_-ZEN was dissolved in 500 μL of methanol before adding 60 μL of 2% GA to obtain the hapten activation solution. The process was performed at room temperature for 4 h with continuous stirring. Hapten activation solution was added dropwise to 20 mg (0.0003 mM) of BSA dissolved in 1.0 mL of 0.1 M, pH 6.7 phosphate buffer (PB) (comprising 38.7 mM NaH_2_PO_4_▪2H_2_O and 61.28 mM Na_2_HPO_4_▪12H_2_O), and then the hapten activation solution was added dropwise to above BSA solution at room temperature for 4 h. The reaction mixture was dialyzed with PBS at 4 °C for three days, and the resulting ZEN-BSA conjugate was stored at 4 °C until further use. The schematic flow for ZEN-BSA (AGA) synthesis is shown in Figure 11.

The synthetic mycotoxin-protein conjugates were confirmed using a DU-800 UV-visible spectrophotometer, and the molecular binding ratios of AFB1 to BSA or OVA, or ZEN to BSA or OVA were calculated based on Lambert–Beer law [43]. Notably, AFB1-BSA (OAE) and ZEN-BSA (AGA) were used for inducing mAbs production, AFB1-OVA (OAE) and ZEN-OVA (AGA) were the coating antigens in an inELISA and an icELISA, and AFB1-BSA (using six different methods, including OAE, MOA, MA, SA, EP, and EED) and ZEN-BSA (using four different methods, including OAE, FA, BDE, and AGA) were the candidate capture antigens on the dual test strip.

### 3.4. Preparation and Assessment of Mycotoxin mAbs

Ten Balb/c mice were randomly divided into two groups, with five mice in each group. The mice were injected subcutaneously on the back at multiple sites (four to six) with 30 μg/head of either AFB1-BSA (OAE) or ZEN-BSA (AGA). The mice received immunogen jabs at an interval of four weeks. For the primary immunization, the emulsion injected was a preparation of immunogen and FCA (1:1, *v*/*v*). For the four subsequent booster immunizations, the FCA in the emulsion was substituted with FIA instead. Blood was collected from the tail of each mouse to extract pAb against AFB1 and ZEN. The titers (representing the immunoreactivity), IC50 (representing the sensitivity), and CR (representing the specificity) of the pAbs were assessed using an inELISA and an icELISA. The mouse with the highest pAb titer, the lowest IC50, and the lowest CR in each group was sacrificed for cell fusion.

The cell fusion of myeloma cells with spleen cells harvested from the mycotoxin-challenged animals was performed as described previously [44]. Ten to fourteen days after cell fusion, the positive clones of interest were screened using inELISA and icELISA. Positive hybridomas were selected for cloning. The mAbs, purified using the saturated ammonium sulfate precipitation method [45], were generated using in vivo-induced ascites method [46].

The IC50 values of the mAbs were determined using icELISA [47]. Briefly, AFB1 (or ZEN) standard stock solution was diluted with 30% methanol-PBS (30:70, *v*/*v*) to varied concentrations (0.33, 1.0, 3.0, 9.0, 27.0, 81.0, and 243.0 µg/L). A corresponding icELISA standard curve was generated by plotting the gradient concentrations (Log C) versus the inhibition percentages, calculated using the following equation: Inhibition percentage (%) = (1 − B/B0) × 100. (Where B is the absorbance of the different concentration standards, B0 is the absorbance of the zero standards). The CR (representing specificities) of the mAbs against mycotoxin was determined using an icELISA as previously described [47]. CR (%) = [IC50 (AFB1 or ZEN)/IC50 (competitors)] × 100%. The mAb at a high titer, highly sensitive (low IC50), and highly specific (low CR) was filtrated for the development of the dual test strip.

### 3.5. Preparation and Identification of AuNPs

AuNPs were prepared using the sodium citrate reduction method as described by Qin et al. [48] but with some modifications. Briefly, 2.0 mL of 1.0% (*w*/*v*) HAuCl_4_ solution was added to 198 mL of boiling ultrapure water in a 250 mL Erlenmeyer flask under constant stirring and configured a total volume of 200 mL and a final concentration of 0.01% (*w*/*v*) HAuCl_4_ solution. After boiling for 5 min, 6.5 mL of 1.0% (*w*/*v*) Na_3_C_6_H_5_O_7_ • 2H_2_O solution was added dropwise to the solution with constant stirring for 15 min. The solution was left to cool before adding ultrapure water to the initial volume. NaN_3_ (0.05% (*w*/*v*)) was then added to the mixture and stored at 4 °C. The AuNPs were characterized using UV and TEM.

### 3.6. Preparation of AuNPs-Labeled mAb

AuNPs-labeled mAbs were prepared using the chessboard method as previously described by Ji et al. [49] but with slight modification. Firstly, the optimal pH for binding of AFB1 (or ZEN mAb) on AuNPs was determined. Here, 1.0 mL of the prepared AuNPs solution was added into eight Eppendorf (EP) tubes (1.5 mL). The pH of the solution was adjusted to 5.5, 6.0, 6.5, 7.0, 7.5, 8.0, 8.5 and 9.0 using 0.25 M K_2_CO_3_ solution. Thereafter, 20 μg of AFB1 mAb (or ZEN mAb) was added to each tube, followed by a 4 °C incubation for 30 min. NaCl (100 μL, 10%) was then added to each tube and incubated at 4 °C for 1 h. The mixtures were centrifuged at 1600× *g* for 30 min at 4 °C and evaluated using UV. The pH at which the highest absorbance occurred was the optimal pH. The optimal amount of AuNPs for labeling mAb was also determined. Briefly, 0.5 mL of AuNPs solution at the optimal pH was added to eight Eppendorf (EP) tubes (1.5 mL). AFB1 mAb (or ZEN mAb) (0.1 mL, 1.0 mg/mL) was added to the first tube and then double-diluted to the eighth tube. The mAb contents were 50, 25, 12.5, 6.25, 3.125, 1.5625, 0.7825, and 0.391 μg, respectively. After thoroughly mixing, the tubes were left to stand for 30 min. NaCl solution (0.1 mL, 10%) was then added, thoroughly mixed, and let to stand for 1 h. The mixtures were centrifuged at 1600× *g* for 30 min at 4 °C and analyzed using UV. The minimum amount of mAb required for 1.0 mL of AuNPs to reach the equilibrium point of the scanning curve and an increase of 10% on this basis was the optimal amount for labeling mAb. Finally, AuNPs-labelled mAb was prepared and purified. Briefly, 20 mL of AuNPs solution at the optimal pH and optimal amount of AFB1 mAb (or ZEN mAb) were mixed in a centrifuge tube and incubated at room temperature for 30 min, and then 2.0 mL of 10% BSA (*w*/*v*) solution was added. After following incubation for another 30 min, the mixture was centrifuged at 8000× *g* at 4 °C for 30 min. The supernatant was discarded, and the precipitate was resuspended with AuNPs resuspension (0.05 M, pH 7.4 borate buffer solution (BBS) comprising Na_2_B_4_O_7_▪10H_2_O 50 mM) containing BSA 0.15 mM, sucrose 87.72 mM, NaN_3_ 0.05 mM). Centrifugation was repeated once. The resuspended solution was filtered using a 0.45 μm membrane and stored at 4°C until further use.

### 3.7. Optimization of the Technical Parameters of a Dual Test Strip

The analytical performance of the dual test strip is affected by numerous parameters, including types and working concentrations of the immunoreagents, as well as types of materials and buffers. Optimization regarded the best working concentration of the coated antigen and AuNPs-labeled mAb. GaMIgG at a concentration of 1.0 mg/mL was prepared using the coating antigen buffer and immobilized on the C line of the NC membrane. Homologous coating antigen AFB1-BSA (OAE) of AFB1 mAb was diluted to a concentration of 2.0, 1.0, 0.5, 0.25, 0.125, 0.0625 mg/mL, and coated on the T1 line of the NC membrane. ZEN-BSA (AGA) was coated on the T2 line of the NC membrane in the same way. The AFB1 mAb for AuNP-labeled was diluted at a ratio of 1:1, 1:2, 1:4, 1:8, 1:16, and 1:32 with AuNPs-labeled mAb dilution buffer and immobilized on the conjugate pad. A similar dilution of AuNP-labeled ZEN mAb was immobilized on the conjugate pad. The optimal working concentration combination of coated antigen and AuNPs-labeled mAb was based on the visual LOD using the chessboard method. Similarly, the five heterologous coating antigens of AFB1 mAb were determined using the working concentration of the screened homologous coating antigen AFB1-BSA (OAE) (or ZEN-BSA (AGA)) as a reference, and based on the visual LOD. The five heterologous coating antigens of AFB1 mAb were determined. The optimal working concentration combination was screened with three heterologous coating antigens of ZEN mAb. Six commonly used NC membranes (Millipore 135, Millipore 180, Prima 40, Prima 85, HF 135, and HF 180) were tested, and the most suitable type of NC membrane was selected by comparing the flow rate, color intensity, and clarity. Four commonly used conjugate pads (SB06, SB08, 8964, and 6613) were tested, and the most suitable conjugate and sample pads were selected based on the stability of different conjugate pads after drying, the release efficiency of AuNPs-labeled antibodies, and the residue of AuNPs. The buffer systems used in this study were selected based on the reports of Zhang et al. [50] and Shim et al. [20].

### 3.8. Assembly of the Dual Test Strip

Figure 12a shows the C line, T1 line, and T2 line on the NC membrane were sequentially coated with GaMIgG, AFB1-BSA, and ZEN-BSA, respectively. The conjugate pad was cured with AuNPs-labeled AFB1 mAb and AuNPs-labeled ZEN mAb mixed in equal volumes. The treated sample pad, conjugate pad, NC membrane, and absorbent pad were coated on the backing plate and cut into thin 0.5 cm test strips. AFB1/ZEN standard solution or sample treatment solution (200 μL) were added to the microwell. A dual test strip was immersed in the microwell solution, and the results were observed in 10 min (Figure 12b).

### 3.9. Validation of the Dual Test Strip

Several main performance indicators of AFB1 and ZEN dual test strip, including sensitivity, specificity (same as its mAb), repeatability, validity period, and reliability, were verified. The LOD, representing sensitivity, was determined using visual examination and the test strip reader method. For the visual method, AFB1/ZEN solutions at a final concentration of 0/0, 0.25/1.25, 0.5/2.5, 1.0/5.0, 2.0/10.0, 4.0/20.0, 8.0/40.0, 16.0/80.0, 32.0/160.0, 64.0/320.0 μg/L were prepared using the sample treatment solution (70% methanol-PBS (7:3, *v*/*v*)). Thereafter, 200 μL of the mixture was added to each microwell, and the two toxins were detected simultaneously using the dual test strip. Each concentration was repeated six times to determine the LOD of the dual test strips. For the test strip reader method, after coloration for 10 min, the T1 and T2 lines were screened using BioDot-TSR 3000 test strip reader. The relative optical density values of the scanning area (G/D × A − ROD (pixel)) were recorded (where G is the graph, D is the density, A is the scanning area, D×A is the optical density of the scanning area, and ROD is the relative optical density). The percentage of relative optical density (G/D × A − ROD (pixel)%) of different concentration standards was the ordinate, and the common logarithmic value of the concentration of different standards was the abscissa. In general, the standard curve of the dual test strip was obtained. According to the Song et al. [51] report and based on the standard curve and regression equation, the LOD of the dual test strip for AFB1 or ZEN was 80% (G/D × A − ROD (pixel)%). The repeatability of the test was evaluated using intra-assay and inter-assay. Six different batches of the dual test strip were used to detect different AFB1/ZEN concentrations (0.5/2.5, 1.0/5.0, 2.0/10.0, 4.0/20.0 µg/L) in maize and pig feeds. Intra-assay variations were measured using six replicates of each spiked sample with a specific concentration of the mycotoxins. Inter-assay variations were based on the test results on the six different days [52]. The validity period of the results was determined by storing the test strips at 4 °C, 25 °C, and 37 °C for different periods. The color changes on days 30, 60, 90, 120, 150, and 180 were then observed [32]. The reliability of the dual test strip was determined by comparing results generated by the test and those of LC-MS/MS. Twenty positive maize meal samples contaminated with AFB1 and ZEN confirmed by LC-MS/MS were obtained from the Agricultural Quality Standards and Testing Technology Research Center of Henan Academy of Agricultural Sciences, and sixty filed samples (15 corn, rice, peanut, and pig feed samples) were obtained from the local market. AFB1 and ZEN detection was performed using dual test strips and LC-MS/MS methods. The reliability of the dual test strip was then evaluated as previously described [53].

### 3.10. Preparation of the Test Sample

Briefly, 5.0 g of dried and ground maize, rice, peanut, and pig feed samples were mixed with 15 mL of 70% methanol-PBS (7:3, *v*/*v*) in a 50 mL centrifuge tube, vortexed for 5 min, and then centrifuged at 1600 g for 5 min. The supernatant was diluted to 50 mL with 0.5% Tween 20-PBS solution (0.5%, *v*/*v*) and used for the dual test strip analysis. All of the analyses were performed in triplicate.

## 4. Conclusions

We developed and optimized a dual lateral flow immunochromatographic assay (dual test strip) for accurate AFB1 and ZEN detection. Two immunogens, AFB1-BSA (OAE) and ZEN-BSA (AGA), were successfully synthesized and linked at molecular binding ratios of 8.64:1 (AFB1:BSA) and 17.2:1 (ZEN:BSA), respectively. The hybridoma cell lines 2A11, 2F6, and 3G2 for AFB1 and hybridoma cell lines 2B6, and 4D9 for ZEN were selected using inELISA and icELISA. The lowest IC50 and CR were observed for AFB1 mAb 2A11 and ZEN mAb 2B6. AuNPs with a particle size of 25 nm were prepared using the sodium citrate reduction method. The optimal pH (6.5) and mAb amount (7.0 μg/mL) for AuNPs-labeled AFB1 mAb and the optimal pH (7.0) and mAb amount (3.5 μg/mL) for AuNPs-labeled ZEN mAb were determined using the chessboard method, respectively. The technical parameters of the dual test strip, such as the types and working concentrations of immunoreagents, types of materials and buffers, were systematically optimized. Under the optimal conditions, the visual LOD of the dual test strip for AFB1 was 1.0 μg/L and 5.0 μg/L for ZEN, but decreased to 0.23 μg/L for AFB1 and 1.53 μg/L for ZEN when using a test strip reader. Further analyses revealed that the dual test strip was highly reproducible. The sensitivity and the specificity of the dual test strip for AFB1 and ZEN detection were comparable to that of LC-MS/MS. The proposed AFB1 and ZEN dual test strip is suitable for rapid and simultaneous detection of AFB1 and ZEN contamination in food and feed samples.

## Figures and Tables

**Figure 1 toxins-14-00615-f001:**
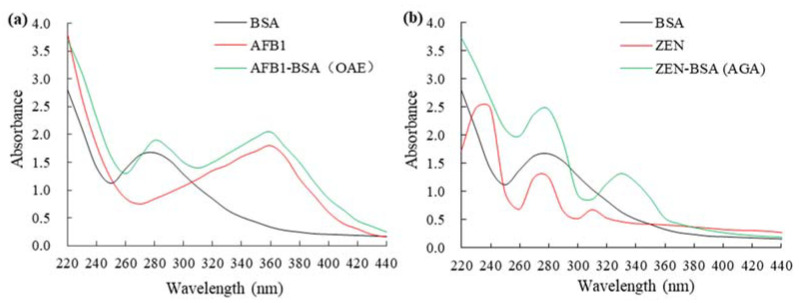
UV spectra of AFB1-BSA and ZEN-BSA: (**a**) UV spectra of AFB1-BSA synthesized via OAE method; (**b**) UV spectra of ZEN-BSA synthesized via AGA method.

**Figure 2 toxins-14-00615-f002:**
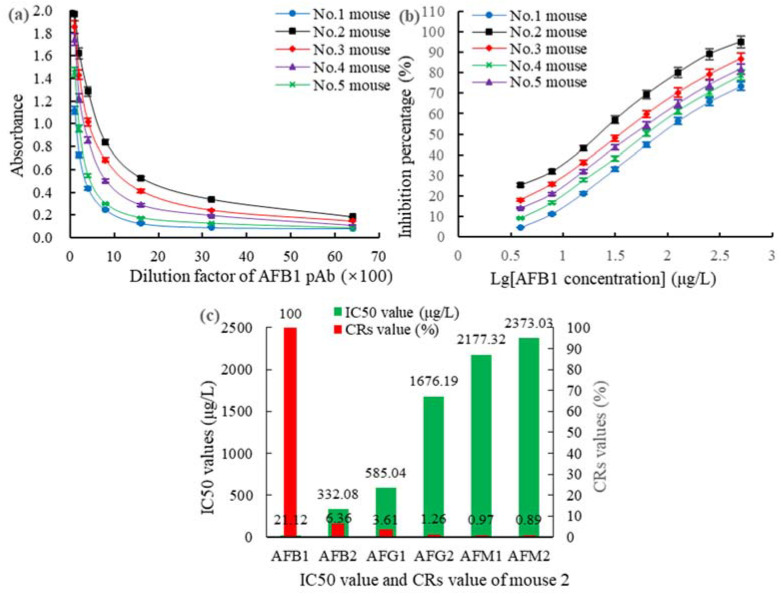
The antibody titers, IC50, and CR% of AFB1 pAb from mice in the AFB1-BSA (OAE) group: (**a**) the AFB1 pAb titers; (**b**) the IC50 of AFB1 pAb; and (**c**) the CR% of AFB1 pAb.

**Figure 3 toxins-14-00615-f003:**
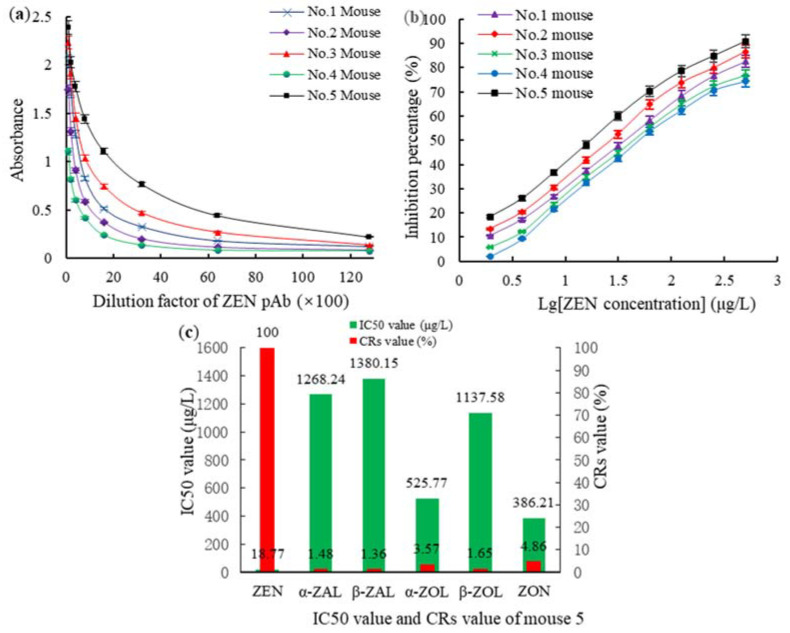
Titers, IC50, and CR% of ZEN pAb derived from ZEN-BSA (OAE): (**a**) Titers; (**b**) IC50 of ZEN pAb; and (**c**) the CR% of ZEN pAb.

**Figure 4 toxins-14-00615-f004:**
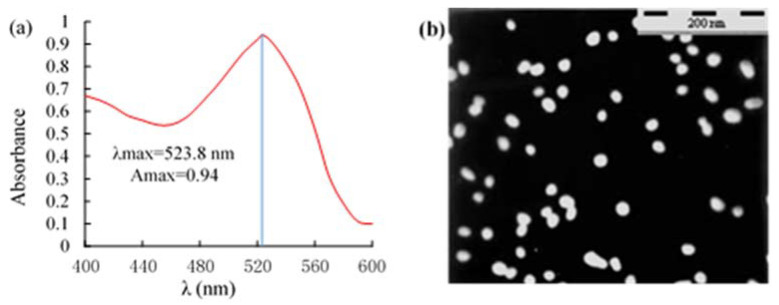
Identification of AuNPs using (**a**) UV; and (**b**) ETM.

**Figure 5 toxins-14-00615-f005:**
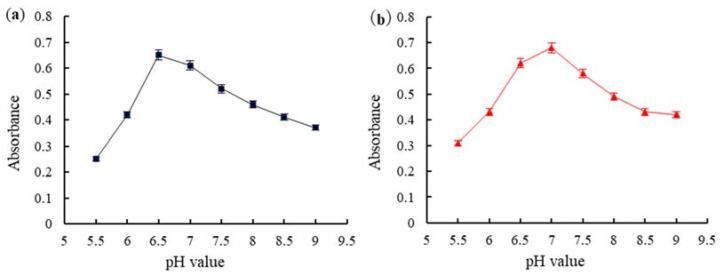
The optimal pH for adsorption of AuNPs on mAbs: (**a**) The effect of pH on the binding between AuNPs and AFB1 mAb; (**b**) The effect of pH on the binding between AuNPs and ZEN mAb.

**Figure 6 toxins-14-00615-f006:**
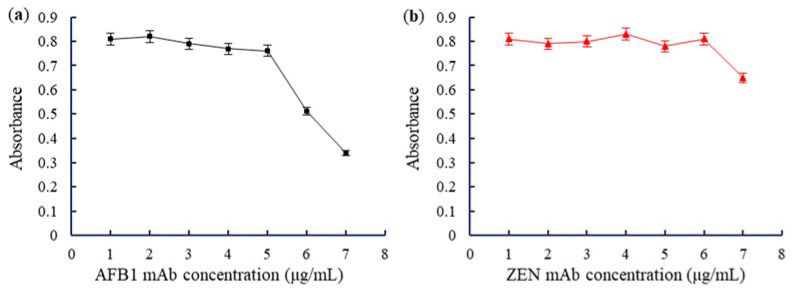
The optimal concentration of AuNPs for labeling mAbs: (**a**) AFB1 mAbs; (**b**) ZEN mAbs.

**Figure 7 toxins-14-00615-f007:**
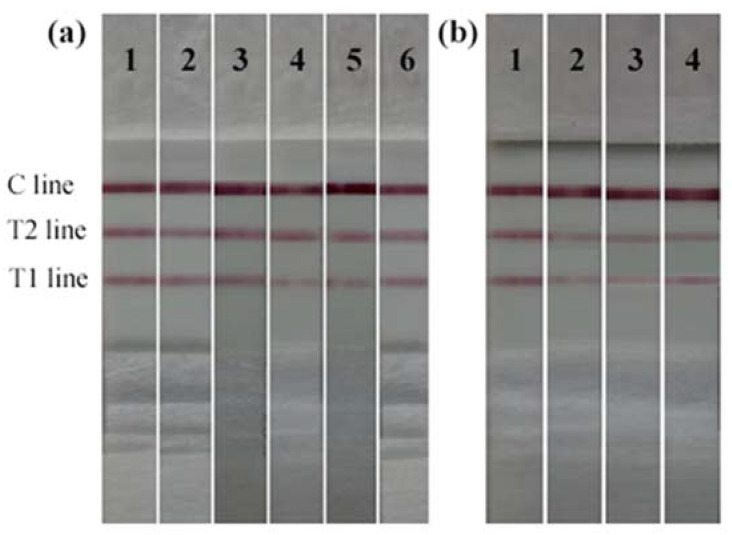
Selection of the (**a**) NC membrane: 1. Millipore 135; 2. Millipore 180; 3. Prima 40; 4. Prima 85; 5. HF 135; 6. HF 180, and (**b**) conjugate pad. 1.6613; 2. SB06; 3. SB08; 4.8964.

**Figure 8 toxins-14-00615-f008:**
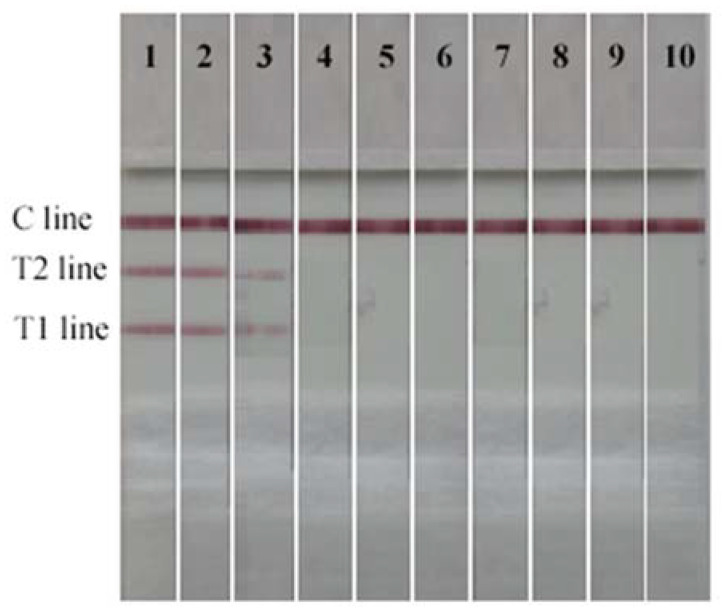
The sensitivity of the dual test strip: 1. AFB1/ZEN, 0/0. 2. AFB1/ZEN, 0.25/1.25. 3. AFB1/ZEN, 0.5/2.5. 4. AFB1/ZEN, 1.0/5.0. 5. AFB1/ZEN, 2.0/10.0. 6.AFB1/ZEN, 4.0/20.0. 7. AFB1/ZEN, 8.0/40.0. 8. AFB1/ZEN, 16.0/80.0. 9. AFB1/ZEN, 32.0/160.0. 10. AFB1/ZEN, 64.0/320.0. (μg/L).

**Figure 9 toxins-14-00615-f009:**
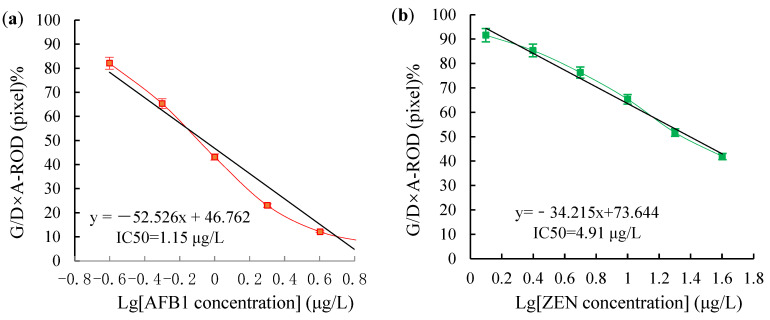
The standard curve of the AFB1 and ZEN dual test strip was detected using BioDot-TSR3000: (**a**) The standard curve of AFB1; (**b**) The standard curve of ZEN.

**Figure 10 toxins-14-00615-f010:**
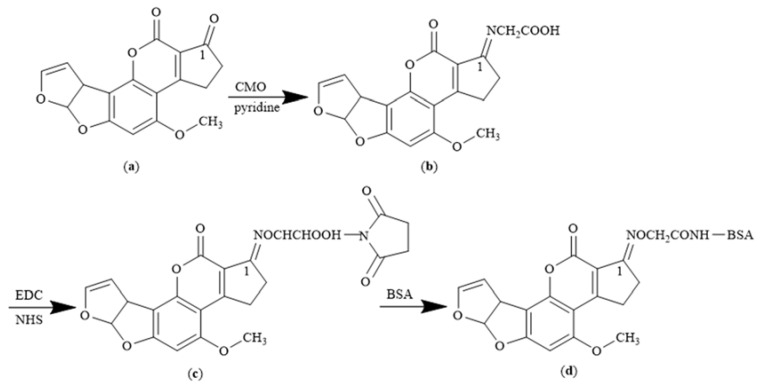
AFB1-BSA synthesis using the OAE method: (**a**) Aflatoxin B1 (AFB1); (**b**) AFB1O; (**c**) Intermediate product; (**d**) AFB1-BSA.

**Figure 11 toxins-14-00615-f011:**
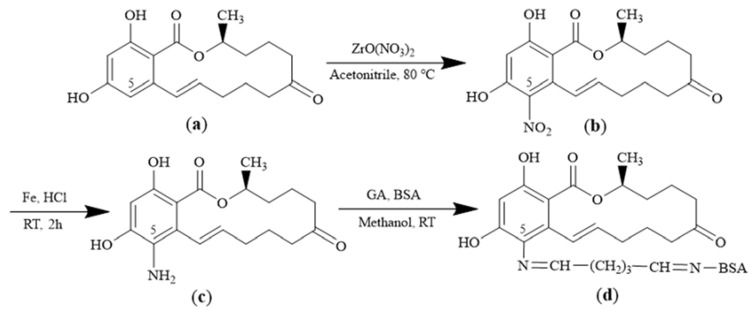
Methods for ZEN-BSA synthesis using the AGA method: (**a**) zearalenone (ZEN); (**b**) 5-NO2-ZEN (AGA); (**c**) 5-NH2-ZEN; (**d**) ZEN-BSA.

**Figure 12 toxins-14-00615-f012:**
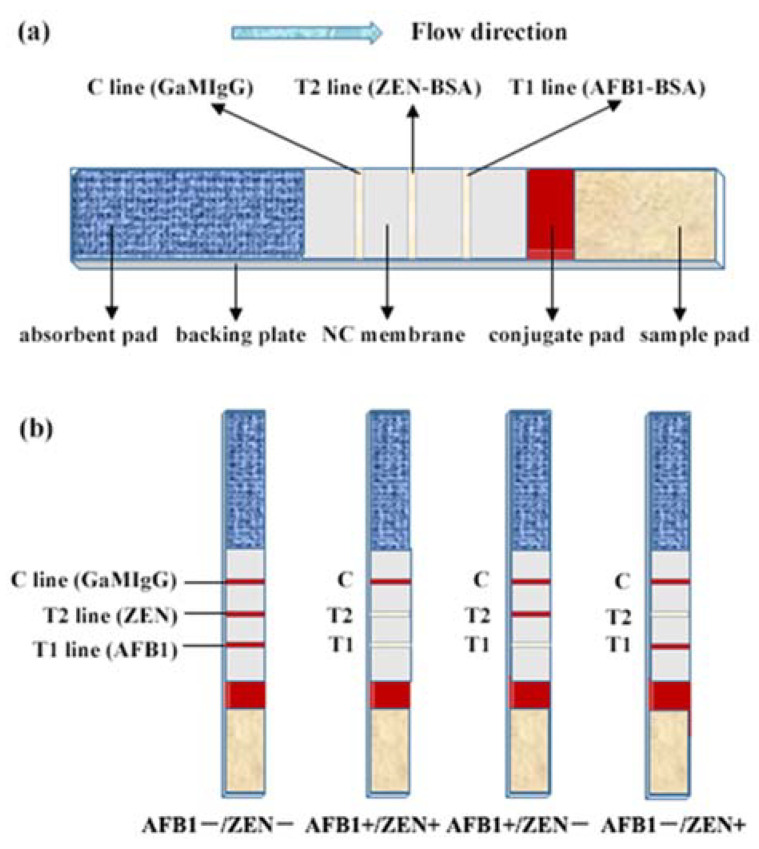
The schematic diagram of the dual test strip for AFB1 and ZEN: (**a**) Schematic diagram of structure; (**b**) Schematic diagram of result judgment. Note: C, control line; T1, AFB1 test line; T2, ZEN test line; +, positive; −, negative.

**Table 1 toxins-14-00615-t001:** The IC50 and CRs of the three AFB1 mAbs against AFB1 analogs.

Compounds	2A11		2F6		3G2	
	IC50 (μg/L) a	CR (%) b	IC50 (μg/L) a	CR (%) b	IC50 (μg/L) a	CR (%) b
AFB1	6.28 ± 0.41	100	7.85 ± 0.57	100	14.36 ± 1.15	100
AFB2	144.32 ± 11.82	4.35	168.82 ± 13.17	4.65	263.97 ± 17.92	5.44
AFG1	272.81 ± 18.22	2.30	310.28 ± 21.42	2.53	469.28 ± 31.08	3.06
AFG2	690.11 ± 44.86	0.91	801.02 ±54.44	0.98	1495.83 ± 95.72	0.96
AFM1	>5000	<0.1	>5000	<0.1	>5000	<0.1
AFM2	>5000	<0.1	>5000	<0.1	>5000	<0.1
Zearalenone	>10,000	<0.1	>10,000	<0.1	>10,000	<0.1
Deoxynivalenol	>10,000	<0.1	>10,000	<0.1	>10,000	<0.1
T-2 toxin	>10,000	<0.1	>10,000	<0.1	>10,000	<0.1
Ochratoxin A	>10,000	<0.1	>10,000	<0.1	>10,000	<0.1

Note: (a) The standard solution was prepared using 70% methanol-PBS (7:3, *v*/*v*). (b) The data were calculated using the CR of AFB1 mAbs against AFB1 as 100%.

**Table 2 toxins-14-00615-t002:** The IC50 and CRs of two ZEN mAbs against ZEN analogs.

Compounds	2B6		4D9	
	IC50 (μg/L) a	CR (%) b	IC50 (μg/L) a	CR (%) b
ZEN	10.38 ± 0.68	100	17.23 ± 1.18	100
α-ZAL	682.89 ± 45.22	1.52	1057.06 ± 65.54	1.63
β-ZAL	810.94 ± 52.72	1.28	1276.30 ± 76.58	1.35
α-ZOL	393.18 ± 23.58	2.64	602.45 ± 3 6.75	2.86
β-ZOL	567.21 ± 39.72	1.83	857.21 ± 57.44	2.01
ZON	243.09 ± 15.81	4.27	353.07 ± 21.88	4.88
Aflatoxin B1	>10,000	<0.1	>10,000	<0.1
Deoxynivalenol	>10,000	<0.1	>10,000	<0.1
T-2 toxin	>10,000	<0.1	>10,000	<0.1
Ochratoxin A	>10,000	<0.1	>10,000	<0.1

Note: (a) The standard solution was prepared using 70% methanol-PBS (7:3, *v*/*v*); (b) The data were calculated using the CR of ZEN mAbs against ZEN as 100%.

**Table 3 toxins-14-00615-t003:** The IC50 and CR of AFB1-specific mAbs reported in previous literature.

References	AFB1 mAb	Coupling Method	Mode	IC50 of AFB1 (μg/L)	CR (%) a				
					AFB2	AFG1	AFG2	AFM1	AFM2
This study (2022)	mAb 2A11	OAE	icELISA b	6.28	4.35	2.30	<1.0	<0.1	<0.1
Jiang et al. (2021) [30]	mAb 1F7	OAE	icELISA	0.15	35.07	8.75	1.15	- c	-
Li et al. (2017) [23]	mAb 1B5	OAE	icELISA	0.012	4.0	3.0	<0.1	<0.1	<0.1
	mAb 2F12	OAE	icELISA	0.01	5.0	2.0	0.2	<0.1	0.2
Zhang et al. (2011) [32]	mAb 3G1	SA	icELISA	1.6	6.4	<0.02	<0.02	-	-
Kolosova et al. (2006) [31]	mAb 34	OAE	dcELISA d	0.62	5.0	31.0	2.4	-	-

Note: (a) The data were calculated using the CR of AFB1 as 100%; (b) icELISA: indirect competitive enzyme-linked immunosorbent assay; (c) -: no detection; (d) dcELISA: direct competitive enzyme-linked immunosorbent assay; SA: semi-acetal.

**Table 4 toxins-14-00615-t004:** The IC50 and CR of ZEN-specific antibodies reported in previous literature.

References	ZEN mAb	Coupling Method	Mode	IC50 of ZEN (μg/L)	CR (%) a				
					α-ZAL	β-ZAL	α-ZOL	β-ZOL	ZON
This study	mAb 2B6	AGA	icELISA b	10.38	1.52	1.28	2.64	1.83	4.27
Sun et al. (2014) [34]	mAb 4A3	BDE	icELISA	1.115	3.854	1.709	2.499	2.800	53.121
Burmistrova et al. (2009) [35]	mAb 2D8	OAE	dcELISA c	0.8	69	<1	42	<1	22
Gao et al. (2012) [33]	mAb # d	FA	icELISA	55.72	0.63	0.92	0.65	0.94	1.48
Teshima et al. (1990) [25]	mAb 7-1-144	AGA	icELISA	11.2	<0.1	<0.1	0.9	<0.1	4.0

Note: (a) The data were calculated using the CR values of ZEN as 100%; (b) icELISA: indirect competitive enzyme-linked immunosorbent assay; (c) dcELISA: direct competitive enzyme-linked immunosorbent assay; (d) #: unnamed.

**Table 5 toxins-14-00615-t005:** The optimal combination concentration for AuNPs-labeled AFB1/ZEN mAb and the coat antigen AFB1-BSA/ZEN-BSA.

AuNPs-Labeled AFB1/ZEN mAb	AFB1-BSA (mg/mL)						ZEN-BSA (mg/mL)					
	2.0	1.0	0.5	0.25	0.125	0.0625	2.0	1.0	0.5	0.25	0.125	0.0625
1:1	++	++	+	-	-	-	++	+	-	-	-	-
1:2	++	++	+	-	-	-	++	+	-	-	-	-
1:4	++	++	+	-	-	-	++	+	-	-	-	-
1:8	++	+	-	-	-	-	+	+	-	-	-	-
1:16	+	-	-	-	-	-	+	-	-	-	-	-
1:32	-	-	-	-	-	-	-	-	-	-	-	-

Note: “++”: clear red line. “+”: light red line. “-” no red line.

**Table 6 toxins-14-00615-t006:** The optimal type combination for AuNPs-labeled AFB1/ZEN mAb and coating antigen AFB1-BSA/ZEN-BSA.

AuNPs-Labeled AFB1/ZEN mAb	AFB1-BSA (1.0 mg/mL)						ZEN-BSA (2.0 mg/mL)				
	OAE	MOA	MA	SA	EP	EED	AGA	OAE	CMA	FA	BDE
1:4	++	+	+	+	-	-	++	+	+	+	+

Note: “++”: clear red line. “+”: light red line. “-”: no red line.

**Table 7 toxins-14-00615-t007:** Scan data of the AFB1 and ZEN T line of the dual test strip. The analysis was performed using Bio Dot-TSR3000.

Toxin Concentration (μg/L)		Lg[Toxin Concentration]		G/D×A − ROD(pixel)		G/D×A − ROD(pixel)%	
AFB1	ZEN	AFB1	ZEN	AFB1	ZEN	AFB1	ZEN
0	0	-	-	16.65	15.52	100	100
0.25	1.25	−0.602	0.099	13.24	14.22	82.60	91.61
0.5	2.5	−0.301	0.398	10.12	13.25	65.78	85.36
1.0	5.0	0	0.699	7.25	11.85	43.54	76.33
2.0	10.0	0.301	1.0	4.31	10.14	25.89	65.36
4.0	20.0	0.602	1.301	2.06	8.02	12.37	51.67
8.0	40.0	0.903	1.602	1.26	6.50	7.57	41.88

**Table 8 toxins-14-00615-t008:** The reproducibility of the dual test strips in different batches.

Batches	The Concentration of Spiked AFB1/ZEN in Corn Samples (μg/L) a				The Concentration of Spiked AFB1/ZEN in Pig Compound Feed Samples (μg/L) a			
	0.5/2.5	1.0/5.0	2.0/10.0	4.0/20.0	0.5/2.5	1.0/5.0	2.0/10.0	4.0/20.0
220210	−b	+ c	+	+	−	+	+	+
220225	−	+	+	+	−	+	+	+
220308	−	+	+	+	−	+	+	+
220316	−	+	+	+	−	+	+	+
220330	−	+	+	+	−	+	+	+
220412	−	+	+	+	−	+	+	+

Note: (a) The readings represent the mean of six replicas. (b) −: negative result. (c) +: positive result.

**Table 9 toxins-14-00615-t009:** The validity period of the dual test strip results.

Time (d)	25 °C				4 °C				37 °C			
	False Negative (%)	False Positive (%)	T1 Color	T2 Color	False Negative (%)	False Positive (%)	T1 Color	T2 Color	False Negative (%)	False Positive (%)	T1 Color	T1 Color
30	0	0	****	****	0	0	****	****	0	0	****	****
60	0	0	****	****	0	0	****	****	0	0	****	****
90	0	0	****	****	0	0	****	****	0	0	****	****
120	0	0	****	****	0	0	****	****	0	0	****	****
150	0	0	****	****	0	0	****	****	2	0	***	***
180	0	0	****	****	0	0	****	****	5	0	**	**

Note: "*" indicates the intensity of the color, the more *, the deeper the color, and the color range is one to four.

**Table 10 toxins-14-00615-t010:** The comparison between the dual test strip and LC-MS/MS in detecting AFB1 and ZEN. A total of 20 samples were analyzed.

Sample Number	Dual Test Strip						LC-MS/MS					
	AFB1			ZEN			AFB1			ZEN		
	Result	PN	PV (μg/L)	Result	PN	PV (μg/L)	Result	PN	PV (μg/L)	Result	PN	PV (μg/L)
20	+	12	≥1.0	+	8	≥5.0	+	12	1.0–11.9	+	8	5.0–367.6

Note: PN: positive number; PV: means positive value.

**Table 11 toxins-14-00615-t011:** Comparison between the dual test strip and HPLC-MS/MS tests in detecting AFB1 and ZEN in different samples.

Sample Type	Sample Number	The Dual Test Strip						LC-MS/MS					
		AFB1			ZEN			AFB1			ZEN		
		Results	PN	PV (μg/L)	Results	PN	PV (μg/L)	Results	PN	PV (μg/L)	Results	PN	PV (μg/L)
Maize	15	+	9	≥1.0	+	7	≥5.0	+	9	1.0–12.5	+	7	5.0–410.7
Rice	15	+	3	≥1.0	+	2	≥5.0	+	3	1.0–9.4	+	2	5.0–235.4
Peanut	15	+	2	≥1.0	+	2	≥5.0	+	2	1.0–6.6	+	2	5.0–176.3
Feed	15	+	8	≥1.0	+	6	≥5.0	+	8	1.0–14.3	+	6	5.0–510.6
Total	60		22			17			22			17	

Note: “PN” means positive number. “PV” means positive value.

## Data Availability

Not applicable.

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
