# Peer review of "A Novel Lateral Flow Immunochromatographic Assay for Rapid and Simultaneous Detection of Aflatoxin B1 and Zearalenone in Food and Feed Samples Based on Highly Sensitive and Specific Monoclonal Antibodies"

_toxins, 2022, doi:10.3390/toxins14090615_

Round 1

Reviewer 1 Report

Dear editor,

In this work authors have developed highly immunoreactive, sensitive, and specific mAbs against AFB1 and ZEN. Then, AuNPs were coupled with the mAbs and a test strip for the simultaneous detection of AFB1 and ZEN was elaborated. In addition, the specificity and sensitivity of the test were confirmed by HPLC-MS/MS. The work appears to be comprehensive; the introductory section provides sufficient background with up-to-date references. The materials and methods section is adequate and allows the authors to achieve their results. The results are clear and the conclusion is based on the findings. All these factors make me believe that this MS is suitable for publication after some corrections.

Major comments:

Table 1 and 2: Please, add information about SD or SE and a statistical comparision among the different mAbs. For instance, in the case of AFB1 mAbs, 2A11 was better, but I believe that a statistical comparison could make your results clearer and interesting.

Minor comments: 

Line 37: Fusarium, Aspergillus, and Penicillium in Italics.

line 42: Aspergillus flavus and Aspergillus parasiticus

line 58: Because ZEN is the most common, abundant, and toxic mycotoxin, it has become the main target for food quality and safety monitoring globally.

- AFB1 is the most toxic mycotoxin, please rewrite this sentence.

line 240: Add a paragraph.

Line 537: centrifuged at 4000 r/min... Convert Rpm to g-force

Reviewer 2 Report

The following comments and corrections are intended to improve the manuscript.

I consider that you could change some of the keywords and include some such as mycotoxins, agro-products, etc.

The name of the microorganisms must be placed in italic font.

Line 52:  What Fusarium species or spp?

Line 64: ibidem.

Line 180-182: So, is the method used in the present study taking into account Li et al, Neuberger et al and Gilfillan et al?

Line 505-506: Reference requiered.

Reviewer 3 Report

I think the manuscript entitled “A novel lateral flow Immunochromatographic assay for rapid and simultaneous detection of aflatoxin B1 and zearalenone in food and feed samples based on highly sensitive and specific monoclonal antibodies” is very interesting. Manuscript is well written and designed. However, I recommend the manuscript to undergo minor revision before the final submission.

1.      The scientific name of microorganism must be in italics, unlike L42 “Aspergillus flavus and Aspergillus parasiticus”.

2.      I suggest authors briefly discuss the limitations of the assay.

3.      I also recommend discussing the minimal detection levels of these toxins required for its identification using the assay, and also to correlate the minimal bioavailability of these mycotoxins in the environment.

4.      I recommend discussing how this assay is advanced from the already existing assays developed by other researchers in the field.

5.      I strongly recommend rewriting the introduction section of the manuscript. The below mentioned papers are suitable for citation:

Xu et al., Toxins (Basel). 2018 Feb; 10(2): 87.

Hao et al., Food and Agricultural Immunology. 2018: 29:1. 498-510.

Dey et al., Crit Rev Food Sci Nutr. 2022 Apr 21;1-22.
